# Effects of Natural Progesterone and Synthetic Progestin on Germ Layer Gene Expression in a Human Embryoid Body Model

**DOI:** 10.3390/ijms21030769

**Published:** 2020-01-24

**Authors:** Yoon Young Kim, Hoon Kim, Chang Suk Suh, Hung-Ching Liu, Zev Rosenwaks, Seung-Yup Ku

**Affiliations:** 1Department of Obstetrics and Gynecology, Seoul National University College of Medicine, Seoul 03080, Korea; yoonykim@snu.ac.kr (Y.Y.K.); obgyhoon@gmail.com (H.K.); suhcs@snu.ac.kr (C.S.S.); 2Institute of Reproductive Medicine and Population, Medical Research Center, Seoul National University College of Medicine, Seoul 03080, Korea; 3Center for Reproductive Medicine and Infertility, Weill Medical College of Cornell University, New York, NY 10065, USA; helenliuhc@gmail.com (H.-C.L.); zrosenw@med.cornell.edu (Z.R.)

**Keywords:** progesterone, progestin, human embryoid body, mouse embryo, early development

## Abstract

Natural progesterone and synthetic progestin are widely used for the treatment of threatened abortion or in in vitro fertilization (IVF) cycles. This in vitro study aimed to assess whether the treatment with natural progesterone or synthetic progestin influences the germ layer gene expression on the early human embryonic development using human embryonic stem cells (hESCs)-derived embryoid bodies (hEBs) as a surrogate of early stage human embryonic development. Human EBs derived from hESCs were cultured for nine days, and were treated with natural progesterone (P4) or synthetic progestin, medroxyprogesterone acetate (MPA) at 10–7 M for five days. To reverse the effects of treatment, mifepristone (RU486) as progesterone antagonist was added to the hEBs for four days starting one day after the initiation of treatment. Mouse blastocysts (mBLs) were cultured in vitro for 24 h, and P4 or MPA at 10^−7^ M was treated for an additional 24 h. The treated embryos were further transferred onto in vitro cultured endometrial cells to evaluate chorionic gonadotropin (CG) expression. To analyze the effects of P4 or MPA, the expression of differentiation genes representing the three germ layers was investigated, GATA-binding factor 4 (GATA4), α-fetoprotein (AFP), hepatocyte nuclear factor (HNF)-3β, hepatocyte nuclear factor (HNF)-4α (endoderm), Brachyury, cardiac actin (cACT) (mesoderm), and Nestin (ectoderm), using quantitative reverse transcription PCR (qRT-PCR) and immunostaining. Significantly lower expressions of HNF-3β, HNF-4α, Brachyury, and Nestin were observed in MPA-treated hEBs (all *p* < 0.05), which was negated by RU486 treatment. This inhibitory effect of MPA was also observed in mouse embryos. Conclusively, the effects of natural progesterone and synthetic progestin may differ in the germ layer gene expression in the hEB model, which suggests that caution is necessary in the use of progestogen.

## 1. Introduction

Natural progesterone (P_4_), produced from granulosa cells during the luteal phase following ovulation, is an essential hormone for embryo implantation and maintenance of early pregnancy [1]. When P_4_ production is insufficient at the time of implantation during early pregnancy, implantation failure or recurrent miscarriage can occur [2]. P_4_ is widely used for luteal phase support during in vitro fertilization (IVF) cycles and is used as a treatment for threatened abortion or recurrent miscarriage [3,4].

During the IVF cycles, gonadotropin-releasing hormone (GnRH) agonists or antagonists are used for the prevention of premature ovulation, and many granulosa cells are aspirated during the oocyte pick-up procedures. For these reasons, progesterone production may be insufficient and thus P_4_ supplementation is common practice in IVF cycles [5].

Medroxyprogesterone acetate (MPA), a synthetic pregnane-derived progestin with high progestational activity and high oral bioavailability, has been prescribed for the induction of withdrawal bleeding in patients with chronic anovulation [5]. The effects of progestin on early pregnancy and fetal development have been controversial to date. In spontaneous menstrual cycles, endogenous P_4_ levels generally rise after ovulation. In IVF cycles, an early rise of serum P_4_ levels on the day of human chorionic gonadotropin (hCG) administration has been reported to be associated with impaired embryo implantation and reduced live birth rate [6]. Additionally, it has been demonstrated that mouse embryos do not progress to the blastocyst stage when cultured in high concentration P_4_-exposed medium [7] and that MPA treatment is embryo-toxic and teratogenic in rats [8]. In contrast, another study showed that the use of MPA during pregnancy does not affect the long-term growth and development of children [9]. However, there are no controlled data in human pregnancy due to ethical limitations of clinical investigations and the absence of a model mimicking human embryonic development [10].

Human embryonic stem cells (hESCs) are derived from the inner cell mass of human blastocyst stage embryos [11,12]. Human embryoid bodies (hEBs) are differentiated from hESCs [13,14] and can be used as a surrogate model [15,16] since they recapitulate many differentiation processes of early human embryonic development [17,18,19]. Previously, we have reported a feasible use for hEBs as an in vitro early human developmental model [20] and other studies showed that the progestational compound affected the differentiation of hESCs and hEBs [21,22,23,24,25].

In this study, we aimed to elucidate the effects of P_4_ and MPA on the differentiation of in vitro cultured hEBs and to validate these results using the gastrulation stage mouse embryos.

## 2. Results

The design of the experiment is represented in Figure 1. The hEBs derived from hESCs were cultured for nine days in suspension and then treated with P_4_ or MPA at 10^−8^ M, 10^−7^ M, 10^−6^ M, and 10^−5^ M for five days. In P_4_ or MPA-treated samples, RU486 was treated for four days in order to check if the observed P_4_ or MPA-specific effect was negated. Mouse BLs (mBLs) were treated with P_4_ or MPA at 10^−7^ M for 24 h, and the expression of three germ layer genes was assessed.

### 2.1. Expression of Three Germ Layer Genes on Undifferentiated hESCs and Differentiated hEBs on Day 5 and Day 9

To characterize the stage of hESC and hEB differentiation, the expression of pluripotency and three germ layer-specific genes on undifferentiated hESCs (UN hES), day 5 hEBs, and day 9 hEBs was evaluated. *Oct4* (undifferentiated, pluripotency marker) was expressed at the highest level on hESCs, decreased on day 5 hEBs, and further down-regulated on day 9 hEBs. The expression of *Nestin* (ectoderm), α-fetoprotein (*AFP*, early endoderm), *HNF3β*, *HNF4α* (endoderm), and *cACT*, *Brachyury* (mesoderm) was low in hESCs, and increased during differentiation and showed the highest levels on day 9 hEBs (* *p* < 0.05) (Figure 2).

The *AFP* expression was significantly increased on day 5 hEBs and other endoderm genes showed a significant increase on day 9 hEBs. Mesoderm-specific genes also showed a significant up-regulation on day 9 hEBs. Thus, day 9 hEBs were deemed as a surrogate of gastrulating stage human embryos. No significant difference was observed between hES XY and XX lines.

### 2.2. Expression of Progesterone Receptors (PRs) on Day 9 hEBs and mBLs

The expression and localization of PRs on the suspension-cultured hEBs were evaluated and confirmed. The existence of PRs was confirmed on the surface of hEBs (Figure 3Aa) and was also observed at the cross-sectioned hEBs as green fluorescence (Figure 3Ab1–b4). These data confirmed the location of PRs both on the inner and outer regions on day 9 hEBs. PR expression was also confirmed on freshly isolated mBLs (Figure 3B).

### 2.3. Effects of P4 and MPA Concentrations on the AFP Gene Expression of hEBs

Since its expression showed the earliest up-regulation among the evaluated genes (Figure 2), the *AFP* gene was selected to test the effects of various P_4_ or MPA concentrations. The serum P_4_ levels of the secretory phase and early pregnancy are known to be less than 100 nM (10^−7^ M) [26]. After treatment with P_4_ or MPA at 10^−8^ M, 10^−7^ M, 10^−6^ M, and 10^−5^ M, the expression of *AFP* was quantitatively evaluated. Supraphysiological levels such as 10^−6^ M, 10^−5^ M showed significant down-regulation of *AFP* expression (* *p* < 0.05 and ***p* < 0.01, respectively) in P_4_ or MPA-treated groups (Figure 4). Thus, 10^−7^ M was selected as a treatment concentration for the following hEBs and mBLs experiments.

### 2.4. Expression of Three Germ Layer Genes on hEBs After P4 and MPA Treatment at 10^−7^ M

The morphology of hEBs was observed under a phase-contrast microscope after P_4_ or MPA treatment at 10^−7^ M for 5 days. The shape, consistency, and diameter of EB did not differ by P_4_ treatment. However, MPA treatment induced cellular degeneration and the structure became loose (Figure 5A).

Five-day P_4_ treatment at 10^−7^ M did not affect the expression of germ layer markers (Figure 5B). In contrast, five-day MPA treatment at 10^−7^ M down-regulated the expression of all the three tested germ layer genes compared to control (* *p* < 0.05, respectively) (Figure 5C upper panel), which was confirmed by a decrease of positive cell population after immunostaining (Figure 5D). The relative susceptibility of one germ layer over the other germ layers was not observed, i.e., the degree of down-regulation did not differ among the three germ layer genes. To confirm that the altered gene expression after MPA treatment was induced by MPA, RU486 was added from one day after the initiation of MPA treatment. When RU486 was treated together with MPA, the altered expression of three germ layer genes was negated (Figure 5C lower panel).

### 2.5. Expression of Three Germ Layer Genes in mBLs after P4 and MPA Treatment

To test the effects of P_4_ or MPA on early embryonic development, mouse blastocysts (mBLs) were collected, cultured for 24 h in vitro, and regarded as gastrulation stage embryos. After MPA treatment at 10^−7^ M for 24 h, three germ layer gene expression was down-regulated on the in vitro cultured mBLs (* *p* < 0.05, respectively) in comparison to the control set (Figure 6A). This dysregulatory marker expression showed a similar pattern to that observed from hEBs. Staining images of three germ layer markers were demonstrated in Figure 6B. MPA-treated mouse BLs showed decreased expression of endoderm specific marker, HNF3β (Figure 6B).

### 2.6. CG Expression of Embryos Transferred onto Endometrial Cells

To evaluate the CG expression of murine embryos, mBLs were cultured with or without P_4_ or MPA treatment after being transferred onto cultured endometrial cells. mBLs were partially or almost fully hatched during in vitro culture. After 24 h of treatment, the embryos (days post coitum (dpc) 5.5–6.0) were plated on the autologous endometrial cells that were collected and cultured after uterine flushing, and CG expression was evaluated. No remarkable morphological difference was observed between control and P_4_ or MPA-treated groups (Figure 7A). The CG expression was lower in the MPA-treated group (Figure 7B).

## 3. Discussion

### 3.1. Main Findings

A significantly lower expression of embryonic three germ layer genes, such as *Nestin*, *HNF-3β*, *HNF-4α*, *GATA4, cACT*, and *Brachyury*, was observed on the in vitro cultured hEBs after MPA treatment at 10^−7^ M for five days. The down-regulated expression of tested genes was not observed when RU486 was added to MPA treatment. In mouse gastrulation stage embryos, the expression of three germ layer genes was significantly down-regulated after MPA treatment at 10^−7^ M for 24 h, which showed a similar pattern to that observed from hEBs. In contrast, this inhibitory effect was not observed in P_4_-treated groups.

### 3.2. Strengths and Limitations

This study investigated the effects of natural progesterone (P_4_) and synthetic progestin (MPA) on the expression of differentiation markers of hEBs and of early stage mouse embryos. Considering the limited access to the in vivo or in vitro experimental manipulation of human embryos due to ethical restrictions, hEBs can be deemed the best available model to assess the effects of high dose progestin on the expression of early embryonic differentiation marker genes with previous findings of the existence of PR receptor in hESCs [27]. The observed findings on hEBs were confirmed using in vitro cultured gastrulation stage murine embryos. The altered expression of early embryonic differentiation-related genes supports the justification of cautious, individualized use, and dosing of a progestational agent in women undergoing IVF or those with threatened abortion.

There are a few things to consider when interpreting our results. First, control and P_4_ or MPA treatment groups consisted of pooled samples, and the expression pattern of each individual sample may show heterogeneity. Secondly, since hEBs are not human embryos themselves, their differentiation potential and gene expression levels may differ from those of human embryos. RU486 alone could inhibit the formation of blastocyst at a concentration 5 μg/mL and above [28]. Third, the observed findings should be interpreted with caution since the used mouse embryos were from syngenic mating and human reproduction is happening allogenically. Lastly, this is an in vitro study and could, with this approach, have missed the entire machinery that exists in vivo, i.e., the activation of the compensatory mechanism in response to the high doses of hormones. Additionally, the rise of P_4_ or MPA is accompanied by an increase of other hormone levels in vivo, and the combined effects may differ compared to those observed in our present study.

### 3.3. Interpretation

Human embryonic stem cells are isolated from the inner cell mass of pre-implantation embryos [11,29,30,31,32] and the formation of hEBs is an intermediate step of h hEB differentiation [33,34]. On hEBs, three germ layer genes express sequentially, i.e., ectoderm and endoderm gene expression precede that of mesoderm [14,20]. This pattern is similar to the gastrulation of mammalian embryos [35] (Figure 1). We chose *Nestin* as ectoderm, *AFP, GATA4, HNF3β,* and *HNF4α* as endoderm, and *cACT* and *Brachyury* as mesoderm genes [36,37]. When the expression of three germ layer-specific genes on hESCs, day 5 hEBs and day 9 hEBs was evaluated, their expression was low on hESCs, and increased during the differentiation process, showing the highest levels on day 9 hEBs (Figure 2).

At the early gastrulation stage, the embryos demonstrate all three germ layers. The differentiation duration was 14 days from undifferentiated hES to day 9 hEB that expressed all three germ layer markers and were regarded as corresponding to the stage of gastrulating mBLs (dpc 3.5). The in vitro differentiation of hEBs was assumed to take approximately five times as long as that of mBLs. Thus, different P_4_ or MPA treatment duration was applied as such in this experimental design (Figure 1).

The expressions of each germ layer markers were differently affected by MPA treatment. Most of the ectodermal and mesodermal genes showed decreased expression after MPA treatment. However, regarding the endodermal genes, the expression of AFP was not significantly affected, while HNF-3β and GATA4 expression was changed by treatment. While the underlying mechanism is currently unclear, these expression patterns may be due to the heterogeneity of hEBs and rapid progression of developmental stages in mouse embryos. Another possible explanation may arise from characteristic gastrulation stage. During the gastrulation, the endoderm is composed of cells from both definitive endoderm and extraembryonic endoderm [38], which may partly explain differential expression of different endodermal genes.

Regarding PRs, their presence on mBLs was reported by Hou and Gorski [39] and on hESCs and hEBs by Hong et al. [27]. In our experiments, considering the physiological P_4_ concentration of 95 × 10^−9^ M at secretory phase of human menstrual cycle [40], various P_4_ concentrations including presumed supraphysiological levels caused by high dose treatment were used, i.e., 10^−8^ M–10^−5^ M. The MPA concentrations were determined in a similar way, referring to the previously reported serum level of MPA in female users [41].

The effects of sex hormones on ESCs differentiation and proliferation [20,42,43] and those on fertilization and early embryonic development are still controversial [8,44,45,46]. However, their effects on human embryonic development are difficult to demonstrate due to ethical reasons. For this purpose, the applicability of hEB models to the investigation on the effects of medication during early stage pregnancy should be further explored.

In clinical practice, medroxyprogesterone acetate (MPA) is widely used for inducing menses in amenorrheic women. MPA has higher binding affinity to PRs, androgen receptors and glucocorticoid receptors compared to natural P_4_ [47], and shares gene regulation with dihydrotestosterone [48]. For this reason, MPA is not used for the luteal phase support in women undergoing IVF despite its teratogenicity not being reported [49,50]. Regarding murine embryonic development and differentiation, high-concentration MPA demonstrated direct embryo-toxic effects [51]. This is in line with the observed down-regulation in the expression of germ layer markers in our present study. Intriguingly, the differed findings between natural P_4_ or synthetic MPA in our study may be explained by their aforementioned molecular and endocrinological characteristics [52].

In conclusion, our findings suggest that the differentiation of human embryos may be affected by iatrogenic treatment and that hEBs can be used as a surrogate model for the early development of human embryos. Clinical use of natural progesterone and synthetic progestin at various concentrations is suggested to be cautious and individualized. The in vitro effects of these compounds should be confirmed in a setting with combined hormonal milieu that simulates early gestational periods including animal models of similar reproductive physiology to humans [53,54,55,56] as well as other affecting factors such as iatrogenic and genetic influence [57,58].

## 4. Methods

### 4.1. Ethics

Use of human embryonic stem cell lines was approved by Institutional Review Board of Institute of Reproductive Medicine and Population, Medical Research Center, Seoul National University (219932-201307-LR-10-01-1) and animal experiments were approved by Institutional Animal Care and Use Committee (IACUC) of Seoul National University Hospital (15-0016-S1A0) and all the animal experiments were performed according to the ethical guidelines of the Association for Assessment and Accreditation of Laboratory Animal Care (AAALAC).

### 4.2. Human Embryonic Stem Cells (hESCs) Culture

As previously reported [13], undifferentiated human embryonic stem cell line, SNUhES3 (46, XY) was cultured on a mitotically inactivated STO (CRL-1503, ATCC, Manassas, VA, USA) feeder layer and passaged every 7 days by mechanical dissociation [59] under stereo-microscope (Nikon, Tokyo, Japan). The hESCs culture medium was composed of DMEM/F12 (Invitrogen, Waltham, MA, USA), 20% knockout serum replacement (KO-SR, Invitrogen, Carlsbad, CA, USA), 1% nonessential amino acids (Invitrogen, Carlsbad, CA, USA), 50 µg/mL streptomycin (Invitrogen, Carlsbad, CA, USA), 50 U/mL penicillin (Invitrogen, Carlsbad, CA, USA), 0.1 mM β-mercaptoethanol (Sigma-Aldrich, St. Louis, MO, USA), and 4 ng/mL basic fibroblast growth factor (bFGF, Invitrogen, Carlsbad, CA, USA).

### 4.3. Human Embryoid Body (hEB) Formation and Culture

Cultured undifferentiated hESCs were treated with 2 mg/mL of collagenase type IV (Invitrogen, Carlsbad, CA, USA) for 30 min at 37 °C. The detached hESC colonies were collected, centrifuged at 1000 g for 5 min, and washed with phosphate buffered saline (PBS). Then, detached hESC colonies were cultured in suspension and their medium was changed every other day. The hEB culture medium was composed of DMEM/F12, 20% KO-SR, 1% nonessential amino acids, 50 µg/mL streptomycin, 50 U/mL penicillin, and 0.1 mM β-mercaptoethanol (Sigma-Aldrich, St. Louis, MO, USA). For this study, day 2 hEBs were filtered through a 100 µm cell strainer (BD Biosciences, Franklin Lakes, NJ, USA) and was divided for further treatment.

### 4.4. P4 or MPA Treatment

For this study, ninety hEBs were used for each treatment group and a total of 1020 EBs were used for further experiments. As demonstrated in Figure 1, to investigate the effects of P_4_ and MPA on the differentiation of hEBs, 10^−8^ M, 10^−7^ M, 10^−6^ M, and 10^−5^ M of P_4_ or MPA (Sigma-Aldrich, St. Louis, MO, USA) was added to the culture media for 5 days from day 9. To confirm the effects of P_4_ and MPA treatment, antagonist of *p*, 10^−7^ M of mifepristone (RU486, Sigma-Aldrich, St. Louis, MO, USA) was added to hEBs for 4 days from one day after the initiation of P_4_ treatment.

### 4.5. Preparation of Mouse Embryos

In this experiment, C57BL/6, six-week-old female mice were used. Briefly, mice were maintained by in-house breeding on a lighting regime of 14 h light and 10 h darkness with food and water supplied ad libitum. Female mice were selected by visual inspection of the vagina and caged with eight-week-old male mice. On the next morning, copulation plug was evaluated and successful mating was confirmed. After a plug check, nine mice were anesthetized using a combination of 100 mg/kg ketamine combined with 10 mg/kg xylazine (0.1 mL/10 g dose, Sigma-Aldrich, St. Louis, MO, USA), and the uteri were isolated and used for embryo collection. Mice were then sacrificed by cervical dislocation according to the guidance of Department of Experimental Animal Research, Biomedical Research Institute, Seoul National University Hospital.

### 4.6. Isolation and Culture of Mouse Embryo

The uteri were briefly rinsed with warmed Hank’s Balanced Salt Solution (HBSS, Invitrogen, Carlsbad, CA, USA) and the embryos were flushed by irrigation of flushing medium (Origio, Målov, Denmark) into uterine horns. Four to ten embryos were obtained from each mouse and three healthy blastocysts (dpc 3.0-4.0) were selected per mouse, pooled and cultured in BlastGen™ (Origio, Målov, Denmark). After 24 h of culture, blastocysts were treated with 10^−7^ M P_4_ or MPA for another 24 h. The embryos were assigned to each group (control, P_4_ or MPA-treated). Each embryo was mixed with single cell-to-CT™ qRT-PCR kit (Ambion, Waltham, MA, USA) buffer and denatured for qRT-PCR. For immunostaining, embryos were fixed in 4% paraformaldehyde (PFA, Sigma-Aldrich, St. Louis, MO, USA) for 15 min at room temperature and used for further processing.

### 4.7. Transfer of Mouse Embryos onto in Vitro Cultured Endometrial Cells

For the analysis of chorionic gonadotropin (CG) expression, mouse embryos were transferred onto in vitro cultured endometrial cells. Briefly, uterine horns of six-week-old female C57BL/6 mice were collected and washed using warmed HBSS. After chopping with surgical blade, the tissues were digested using 1 mg/mL of collagenase type I (Invitrogen, Carlsbad, CA, USA) for 1 h at 37 °C. Then, digested tissues were filtered through a 70 μm cell strainer (BD Biosciences, San Jose, CA, USA) and collected using centrifugation. The retrieved endometrial cells was replated and cultured for 2 days in the medium consisting of DMEM/F12 with 10% fetal bovine serum (FBS, Invitrogen, Carlsbad, CA, USA). Control and MPA-treated embryos were transferred onto endometrial cells and cultured for 24 h.

### 4.8. Morphological Evaluation and Preparation of Paraffin Section for Human Embryoid Body

Growth of hEBs was observed using a phase-contrast microscope (TE-2000, Nikon, Tokyo, Japan) and pictures were taken using the i-solution imaging program (i-solution, Daejeon, Korea). For hEB paraffin section, hEBs were fixed with 4% paraformaldehyde (PFA, Sigma-Aldrich, St. Louis, MO, USA) for 15 min at room temperature, and then rinsed with PBS. Samples were sequentially processed with water, and 70%, 80%, 90%, and 100% ethanol (EtOH, Sigma-Aldrich, St. Louis, MO, USA) for 30 min each at room temperature. Then, samples were processed in xylene (Sigma-Aldrich, St. Louis, MO, USA) for 30 min at room temperature and samples were soaked with melted paraplast (Sigma-Aldrich, St. Louis, MO, USA) and solidified as paraffin block for microtome section. Solidified paraffin blocks were cut at a thickness 5 µm using microtome (Leika, Wetzlar, Germany) and transferred to microslide. Slides were dried overnight and stored at room temperature until ready for use. For the preparation of deparaffinized and rehydrated tissue slides, slides were placed in a 56 °C oven for 10 min to melt paraffin and deparaffinize the slides in xylene for 5 min. Then, slides were incubated with 100% EtOH for 3 min and transferred to 95% EtOH for 3 min. Finally, slides were rinsed with PBS twice for 5 min each time and were used for further immunostaining.

### 4.9. Immunostaining of hEBs and Mouse Embryos

hEB-deparaffinized slide or fixed mouse embryos were washed with PBS. After rinsing with PBS, embryos were permeabilized in PBST [PBS containing 0.05% tween 20 (Sigma-Aldrich, St. Louis, MO, USA)] for 30 min at room temperature and blocked in blocking butter, which consisted of PBS containing 3% bovine serum albumin (BSA, Sigma-Aldrich, St. Louis, MO, USA) and 0.3% Triton X-100 (Sigma-Aldrich, St. Louis, MO, USA). Primary antibodies, rabbit anti-human Progesterone Receptor (PR) (ab32085, Abcam, Cambridge, MA, USA), mouse anti-PR (ab2765, Abcam, Cambridge, MA, USA), mouse anti-human Nestin (ab22035, Abcam, Cambridge, MA, USA), rabbit anti-human and mouse Brachyury (ab20680, Abcam, Cambridge, MA, USA), goat anti-human HNF3β (AF2400, R&D Systems, Minneapolis, MN, USA), mouse anti-mouse Nestin (ab6142, Abcam, Cambridge, MA, USA), and rabbit anti-mouse HNF3β (ab108422, Abcam, Cambridge, MA, USA) were diluted 1:100 in blocking buffer and samples were incubated at 4 °C overnight. Then, they were rinsed three times in PBST for ~15 min each. Secondary antibodies, Alexa Fluor^TM^ 488 goat anti-rabbit (A11008), Alexa Fluor^TM^ 488 goat anti-mouse (A11001), Alexa Fluor^TM^ 488 rabbit anti-goat (A11078), Alexa Fluor^TM^ 594 rabbit anti-goat (A27016), and Alexa Fluor^TM^ 594 goat anti-mouse (A11005, all purchased from Molecular Probes, Waltham, MA, USA), were diluted 1:500 in blocking buffer and samples were incubated in secondary antibody for 1 h at room temperature. Samples were then rinsed three times in PBST and incubated briefly with 4′,6-diamidino-2-phenylindole, dihydrochloride (DAPI, D1306, Molecular Probes, Waltham, MA, USA) reagent. For negative control, only secondary antibodies were treated to samples, as described above. Fluorescence images were captured using EVOS-FL fluorescence microscope (Thermo Fisher Scientific, Waltham, MA, USA). Specific antibodies used for immunostaining are summarized in Table 1. To measure the positive portion, DAPI and fluorescence-positive cells were counted, and the percentage of positive cells was calculated (Figure 5D).

### 4.10. RNA Isolation and Quantitative Reverse Transcription-Polymerase Chain Reaction (qRT-PCR) of hESCs, hEBs and mBLs

We assessed the expression of eight genes as follows: *Oct4* (undifferentiated state); *Nestin* (ectoderm); α-fetoprotein (*AFP*), *GATA4,* hepatocyte nuclear factor (*HNF*)-*3β,* and *HNF-4α* (endoderm); *cACT* (cardiac actin) and *Brachyury* (mesoderm). Specific primers used for qRT-PCR are shown in Table 2.

For the analysis of gene expression in hESCs and hEBs, one hundred hESC colonies and eighty hEBs were used for total RNAs isolation using Trizol (Invitrogen, Carlsbad, CA, USA). cDNA was synthesized from 0.5 µg of total RNA using AccuPower^®^ RT PreMix (Bioneer, Daejeon, Korea). Quantitative PCR was performed in RotorGene Q (Qiagen, Valencia, CA, USA) using QuantiTect SYBR green PCR kit (Qiagen, Valencia, CA, USA) as uniplex setting. The amplification program included an initial step at 95 °C for 10 min, followed by 30 cycles of denaturation at 95 °C for 15 s, annealing at 58 °C for 20 s, and extension at 72 °C for 30 s. All reactions were run in triplicate and relative gene expression was normalized using the corresponding *β-actin* expression.

For the analysis of gene expression in mBLs, qRT-PCR was performed using single cell-to-CT™ qRT-PCR kit (Ambion, Naugatuck, CT, USA). Collected embryos were rinsed with PBS and mixed with lysis buffer using delicate pipetting. Then, mixed samples were incubated for 5 min at room temperature and was followed by mixing with stop solution for 2 min at room temperature. After lysis and stopping, samples were chilled on ice and mixed with qRT-PCR master mix and run in RotorGene Q as uniplex setting. The amplification program included an initial step at 48 °C for 30 min, 95 °C for 10 min, and followed by 40 cycles of denaturation at 95 °C for 15 s, annealing at 60 °C for 60 s. All reactions were run in triplicate and relative gene expression was normalized using the corresponding *β-actin* expression.

### 4.11. Statistical Analysis

Statistical analysis was performed using SPSS version 21.0 (SPSS Inc., Chicago, IL, USA). Data are expressed as mean ± SD, and were analyzed using ANOVA. *p* value less than 0.05 was considered to be statistically significant.

## Figures and Tables

**Figure 1 ijms-21-00769-f001:**
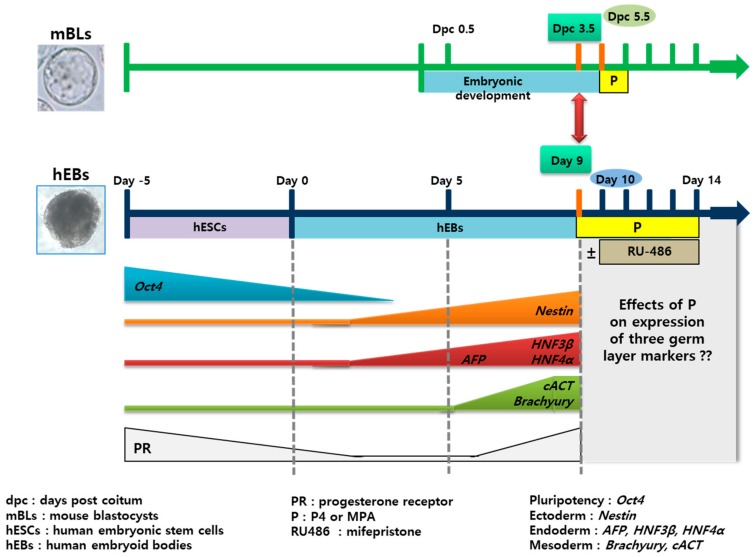
Schematic presentation representing the experiments using human embryoid bodies (hEBs) and mouse embryos. hEBs were formed from human embryonic stem cells (hESCs) using suspension culture. hEBs cultured for 9 days were treated with progesterone (P_4_) or medroxyprogesterone acetate (MPA) at 10^−8^ M, 10^−7^ M, 10^−6^ M, and 10^−5^ M for 5 days. RU486 was treated for 4 days to check if the post-treatment changes in the expression of three germ layer marker genes were P_4_ or MPA-specific. Mouse blastocysts (mBLs) were treated with P_4_ or MPA at 10^−7^ M for 24 h, and the expression of three germ layer marker genes was assessed. The expression level of pluripotency, ectoderm, endoderm, and mesoderm-specific genes and of progesterone receptor (PR) gene is depicted.

**Figure 2 ijms-21-00769-f002:**
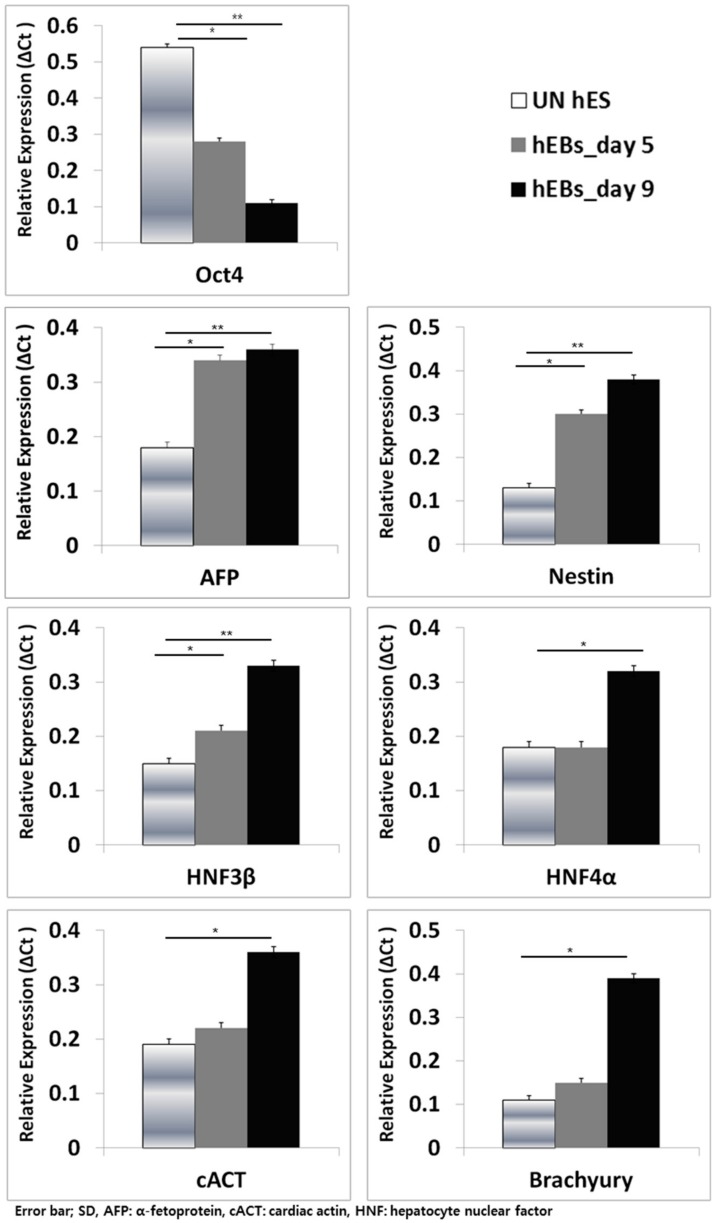
Expression of pluripotency and three germ layer markers on undifferentiated hESCs (UN hES), day 5 hEBs (hEBs-day 5), and day 9 hEBs (hEBs-day 9). *Oct4* (undifferentiated, pluripotency marker) was expressed at the highest level on hESCs and decreased throughout the differentiation stages. The expression of *Nestin* (ectoderm), α-fetoprotein (*AFP*), hepatocyte nuclear factor (*HNF) 3β*, *HNF4α* (endoderm), and cardiac actin *(cACT)*, *Brachyury* (mesoderm) was the highest on day 9 hEBs (* *p* < 0.05 and ** *p* < 0.01, respectively).

**Figure 3 ijms-21-00769-f003:**
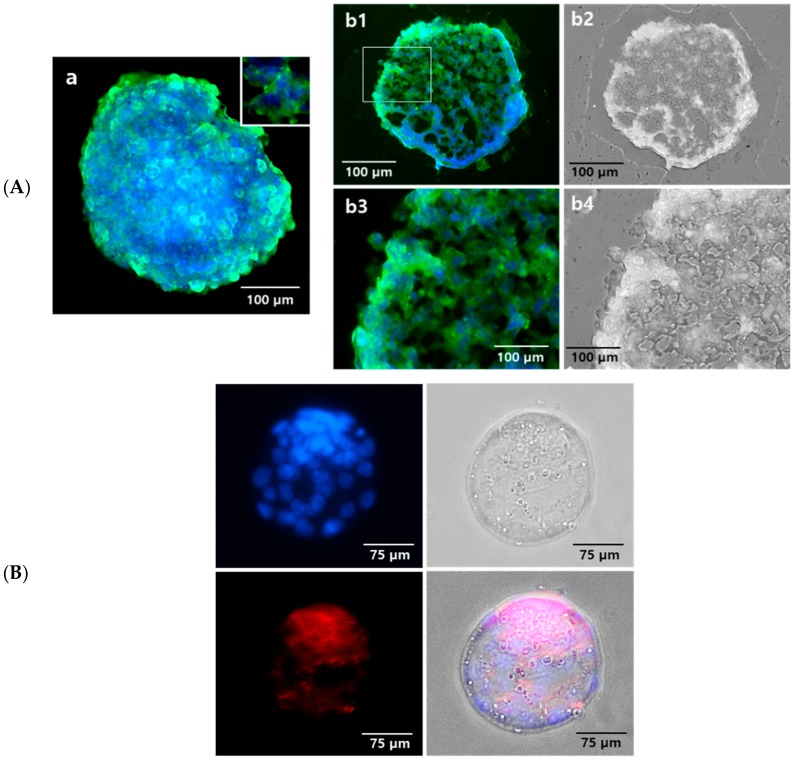
Expression of progesterone receptor (PR) on the day 9 hEBs and mouse BLs. (**A**) Suspended hEBs were fixed and immunostained with PR-specific antibody (green fluorescence). a: PR expression on the surface of hEB. Pictures with higher magnification represented in inner box. b1, b3: PR expression on the cross-sectioned hEB. b2, b4: phase-contrast microscopic image. (**B**) The expression of PR was confirmed in mouse embryo. Red fluorescence represented PR expression, and nuclei were stained with 4′,6-diamidino-2-phenylindole (DAPI) and are represented as blue.

**Figure 4 ijms-21-00769-f004:**
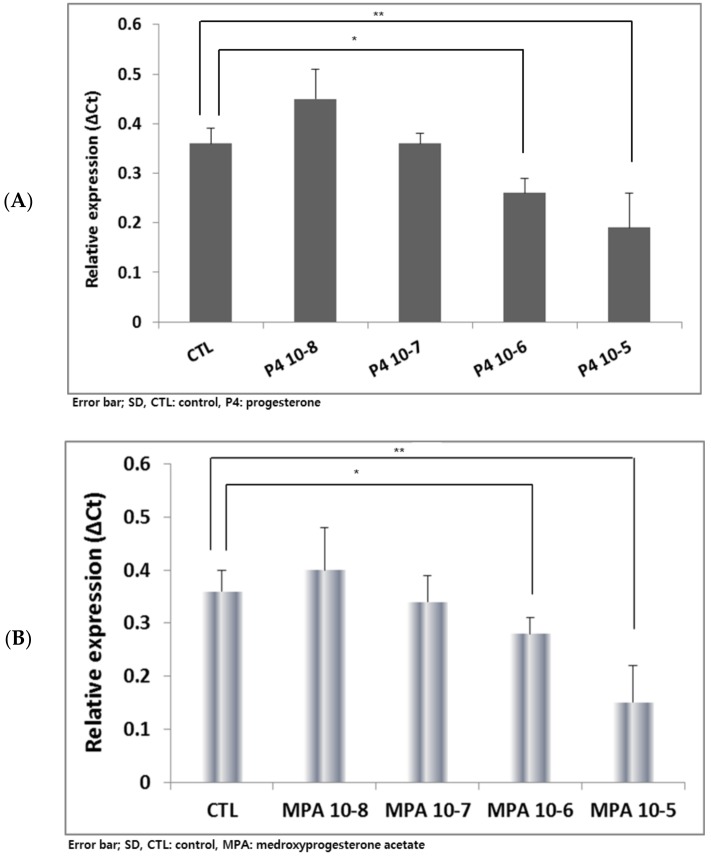
The effects of natural progesterone and synthetic progestin at various concentrations on the expression of α-fetoprotein *(AFP)* of day 9 hEBs. Treatment of progesterone (P_4_) or medroxyprogesterone acetate (MPA) of various concentration ranges affected the expression of *AFPs*, i.e., supraphysiological concentrations such as 10^−6^ M, 10^−5^ M induced a significant down-regulation of *AFP* expression (* *p* < 0.05 and ** *p* < 0.01, respectively) (**A**,**B**).

**Figure 5 ijms-21-00769-f005:**
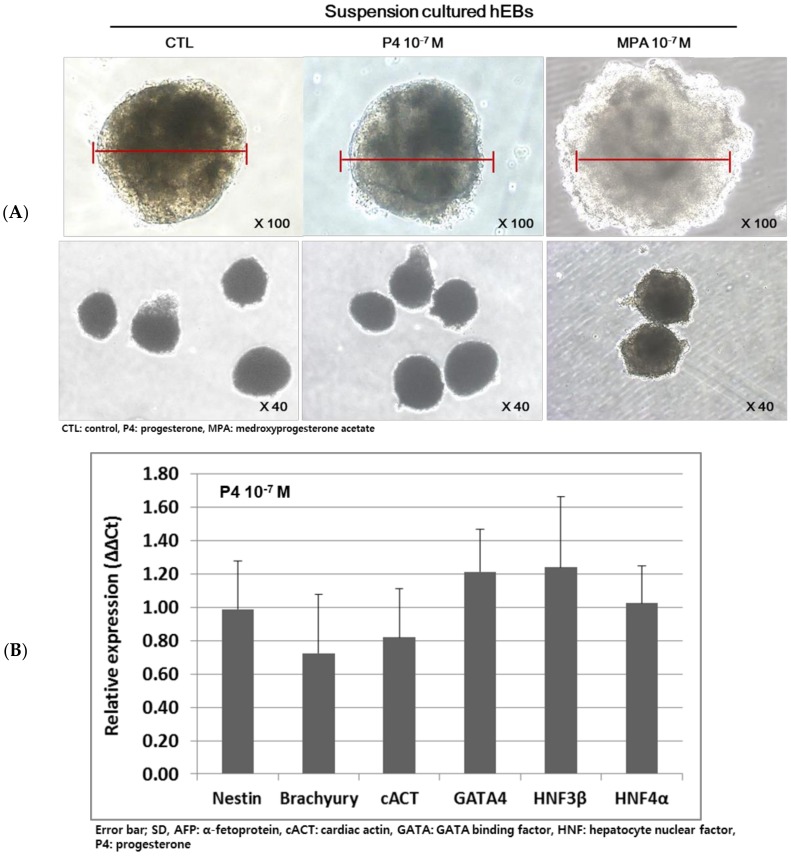
Effects natural progesterone and synthetic progestin on the expression of three germ layer genes in day 9 human embryoid bodies (hEBs). (**A**) The morphology did not differ according to the progesterone (P_4_) treatment at 10^−7^ M for 5 days. The medroxyprogesterone acetate (MPA)-treated hEBs were unformed and became loose. The red bar of each picture indicates the diameter of formed EBs. (**B**) P_4_ treatment at 10^−7^ M for 5 days did not alter the expression of germ layer genes. (**C**) MPA treatment at 10^−7^ M for 5 days induced down-regulated expression of all tested genes (* *p* < 0.05, respectively, CTL: untreated hEBs), upper panel. RU486 treatment negated the effects of MPA (CTL: MPA-treated hEBs), lower panel. (**D**) Immunostaining of three germ layer genes on the hEBs treated with MPA. Counts of positively stained cells were evaluated and compared between the groups (* *p* < 0.05).

**Figure 6 ijms-21-00769-f006:**
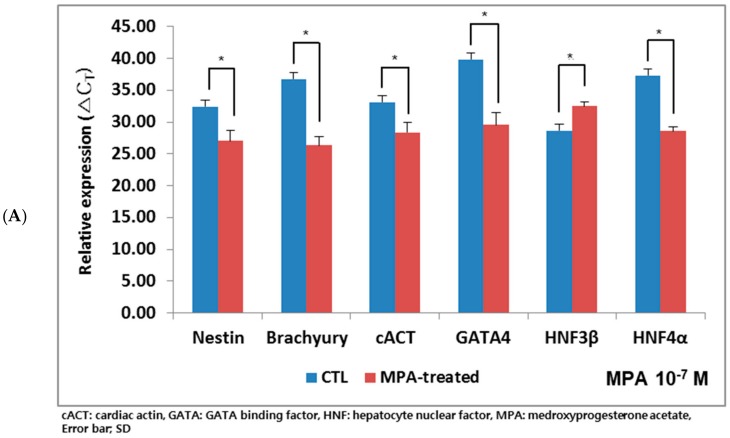
Effects of MPA on the expression of three germ layer genes on mouse embryos. Mouse blastocysts (mBLs) were cultured for 24 h, and treated with MPA at 10^−7^ M for an additional 24 h. (**A**) All tested genes were down-regulated after MPA treatment (* *p* < 0.05, respectively, CTL: MPA-treated mBLs). (**B**) Immunostaining of in vitro cultured mouse embryos (control vs. MPA).

**Figure 7 ijms-21-00769-f007:**
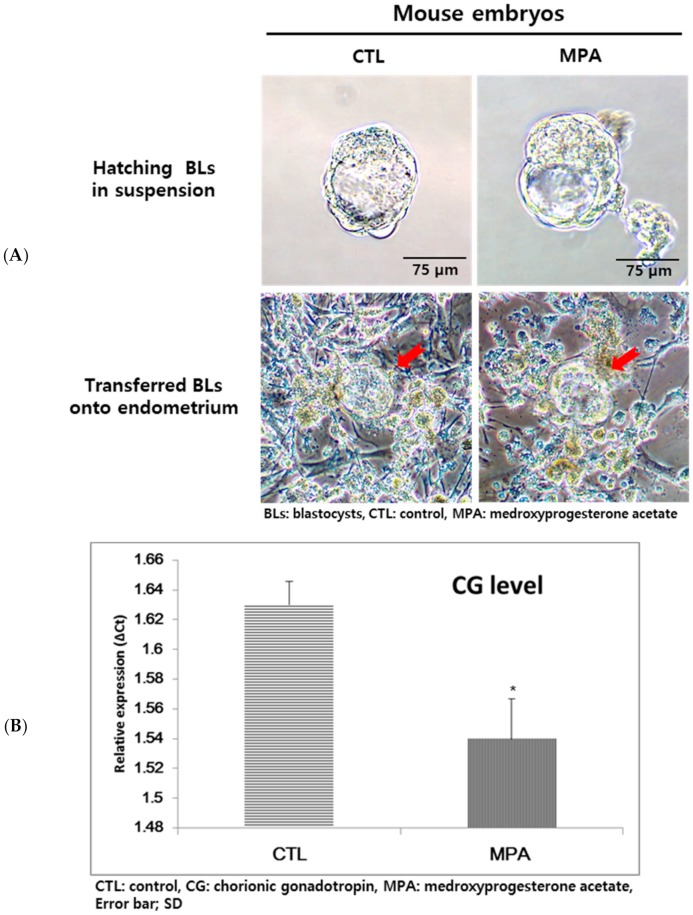
Chorionic gonadotropin (CG) expression of mouse embryos transferred onto autologous endometrial cells. (**A**) Mouse embryos transferred onto primary in vitro cultured endometrial cells did not show any different morphology according to MPA treatment at 10^−7^ M. The red arrow indicated mBLs. (**B**) The expression of CG was down-regulated in the MPA-treated group (* *p* < 0.05).

**Table 1 ijms-21-00769-t001:** Primary antibodies used for this study.

Name	Company	Cat. No.
**Human**		
rabbit anti-human progesterone receptor (PR)	Abcam	ab32085
mouse anti-human Nestin	Abcam	ab22035
rabbit anti-human Brachyury	Abcam	ab20680
goat anti-human HNF3β	R&D Systems	AF2400
**Mouse**		
mouse anti-PR	Abcam	ab2765
mouse anti-mouse Nestin	Abcam	ab6142
rabbit anti-mouse Brachyury	Abcam	ab20680
rabbit anti-mouse HNF3β	Abcam	ab108422

**Table 2 ijms-21-00769-t002:** The sequence of primers used for qRT-PCR.

Gene	Forward	Reverse
**Human**		
*AFP*	AGAACCTGTCACAAGCTGTG	GACAGCAAGCTGAGGATGTC
*HNF3β*	CTACGCCAACATGAACTCCA	GAGGTCCATGATCCACTGGT
*HNF4* *α*	TGTCCCGACAGATCACCTC	CACTCAACGAGAACCAGCAG
*Brachyury*	TAAGGTGGATCTTCAGGTAGC	CATCTCATTGGTGAGCTCCCT
*cACT*	GGAGTTATGGTGGGTATGGGTC	AGTGGTGACAAAGGAGTAGCCA
*GATA4*	TTACACGCTGATGGGACTGGAG	GGGGAACGCAGGGGACAAG
*Nestin*	CAGCTGGCGCACCTCAAGATG	AGGGAAGTTGGGCTCAGGACTGG
*Oct4*	GAGAACAATGAGAACCTTCAGGA	CTCGAACCACATCCTTCTCT
*β-actin*	AGAGCTACGAGCTGCCTGAC	AGCACTGTGTTGGCGTACAG
**Mouse**		
*AFP*	AGTTTCCAGAACCTGCCGAG	ACCTTGTCGTACTGAGCAGC
*CGβ*	GTCAACACCACCATCTGTGC	GGCAGAGTGCACATTGACAG
*HNF3β*	TATTGGCTGCAGCTAAGCGG	GACTCGGACTCAGGTGAGGT
*HNF4α*	CTTCCTTCTTCATGCCAG	ACACGTCCCCATCTGAAG
*Brachyury*	AACTTTCCTCCATGTGCTGAGAC	TGATTCCCAACACAAAAAGCT
*cACT*	GGATTCTGGCGATGGTGTAA	CTCGTTGCCAATGGTGATGAC
*GATA4*	CCTTCGACAGCCCGGTCCT	TGCACAGATAGTGACCCGTCCC
*Nestin*	GGATACAGTTTATTCAAGG	CAGCCGCTGAAGTTCACTCT
*β-actin*	CGCCACCAGTTCGCCATGGA	TACAGCCCGGGGAGCATCGT

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
