# Peer review of "Effects of Natural Progesterone and Synthetic Progestin on Germ Layer Gene Expression in a Human Embryoid Body Model"

_ijms, 2020, doi:10.3390/ijms21030769_

Round 1
Reviewer 1 Report
The manuscripts was wriiten well and presentation of the results are appropriate. However, this manuscript addressed limited implementation not enough for appearing on the journal.
Author Response
The manuscripts was written well and presentation of the results are appropriate. However, this manuscript addressed limited implementation not enough for appearing on the journal.
Reply: The authors appreciate the in-depth comment and agree with the pointed by the reviewer. The demonstrated data did not include further development of hEB or mBL, the in vivo implantation and development.
As addressed throughout the manuscript, this study suggested a possibility of using human embryoid body (hEB) as an early human embryo-like model. Although the data does not exactly corresponds to in vivo environment, our data implied that the hEB was affected by progestins. As indicated in Introduction and Discussion, the Ps are widely used in clinic for the inhibition of miscarriage, however, it is impossible to monitor the early human development for ethical and technical reasons. The current clinical limitation leads the authors to design an in vitro pre-implantation response model of molecules affecting early human embryo development. Despite the different characteristics of hESC lines from embryos, it can still serve as one of the useful models for early development.
Published articles with similar hypothesis demonstrated early development-focused data. Here are the examples of published articles.
Gallego et al., Opioid and progesterone signaling is obligatory for early human embryogenesis. Stem Cells Dev. 2009;18(5):737-40.
Gallego et al., The pregnancy hormones human chorionic gonadotropin and progesterone induce human embryonic stem cell proliferation and differentiation into neuroectodermal rosettes. Stem Cell Res Ther 2010;1:28, doi:10.1186/scrt28
Jeon et al., Altered expression of epithelial mesenchymal transition and pluripotent associated markers by sex steroid hormones in human embryonic stem cells. Mol Med Rep. 2017;16(1):828-36.
Wang et al., A human embryonic stem cell-based model for benzo[a]pyrene-induced embryotoxicity. Reprod Toxicol. 2019;85:26-33.
Reviewer 2 Report
Well written manuscript and address important topic.
1- Please mention what error bars reflect (i.e. SD or SEM) under each figure.
2- Abbreviations should be described when mentioned for first time in the manuscript e.g AAALAC in L72. GATA4....etc.
3-Abbreviations used in figures should be described under each figure we.g HNF3β, HNF4α in figure 2....etc.
4- Statistical analysis, authors are using T-test. I think it should be ANOVA since you have more than 2 comparison followed by Tukey HSD as post-hoc
4- Please, revise statistical analysis results; e.g Figure 2; apparently, AFP should be significantly different in hEBs_day5 in comparison to UN hES. Moreover, may be it is better to use superscripts (a, b, c) to show significant differences
Author Response
COMMENTS: Please mention what error bars reflect (i.e. SD or SEM) under each figure.
REPLY: Referee’s comments surely helped to improv the quality of figures. The error bar indicates SD and mentioned in each revised figure and legends.
COMMENTS: Abbreviations should be described when mentioned for first time in the manuscript e.g AAALAC in L72. GATA4....etc.
REPLY: The authors express sincere thanks for the considerate comments. The following abbreviation’s full name has been provided as it was first mentioned.
GATA4 now revised as GATA-binding factor 4 (GATA4) in it first appeared in the Abstract section. IACUC and AAALAC corrected as Institutional Animal Care and Use Committee (IACUC) and Association for Assessment and Accreditation of Laboratory Animal Care (AAALAC) in it first appeared in the Methods section. Abbreviations in Methods section such as PBS, FBS and Ethanol has been corrected as phosphate buffered saline (PBS), fetal bovine serum (FBS, Invitrogen) and ethanol (EtOH).
COMMENTS: Abbreviations used in figures should be described under each figure we.g HNF3β, HNF4α in figure 2....etc.
REPLY: Once again, the authors appreciate the comments of reviewer. The abbreviation used all indicated at the bottom of each figure.
COMMENTS: Statistical analysis, authors are using T-test. I think it should be ANOVA since you have more than 2 comparison followed by Tukey HSD as post-hoc.
REPLY: The authors appreciated the constructive and in-depth comments of reviewer. The data has been statistically analyzed using ANOVA and revised in Materials and Methods section.
COMMENTS: Please, revise statistical analysis results; e.g Figure 2; apparently, AFP should be significantly different in hEBs_day5 in comparison to UN hES. Moreover, may be it is better to use superscripts (a, b, c) to show significant differences.
REPLY: Reviewer’s comment greatly helped to improve the statistical analysis of the data. The data has been revised accordingly and the figures have been revised.
Reviewer 3 Report
The manuscript of Kim et al., is mainly focused on embryoid body, which considered as model for early human development. Although it is interesting data, major revision is needed for the solidity of data and better understanding of readers.
The comments are
It is interesting data, however, the manuscript needs knowledge of embryo development. Short description of early development of mouse and human surely will increase the understanding of readers. I suggest that the authors answer the following questions: why have you used the male hESC instead for an XX hESC line? I recommend to use XX hESC line. How did you control for the colony size and resulting size of hEBs? How well? What is the variation in sizes - heterogeneity in size may impact on differentiation. It would be of interest to see the effect of progesterone treatment across the time course of treatment- is differentiation being blocked by progesterone or simply delayed? Were unused embryos simply the wrong stage or did they arrest in culture? This paragraph of the methods needs clarification. Were the animal experiments conducted according to the local ethical committee?
Author Response
COMMRNTS: It is interesting data, however, the manuscript needs knowledge of embryo development. Short description of early development of mouse and human surely will increase the understanding of readers.
REPLY: The constructive comment is reflected in Discussion section and the sentences are as follows.
Interpretation
Human embryonic stem cells are isolated from the inner cell mass of pre-implantation embryos [11,30,31] and the formation of hEBs is an intermediate step of hESC differentiation [32,33]. On hEBs, three germ layer genes express sequentially, i.e. ectoderm and endoderm gene expression precede that of mesoderm [14,20]. This pattern is similar to the gastrulation of mammalian embryos [34] (Fig 1).
COMMRNTS:Why have you used the male hESC instead for an XX hESC line? I recommend to use XX hESC line.
REPLY: This is an important issue in many studies using hESC lines. Many researchers prefer to use XY lines since XX lines are considered to show relatively less efficient differentiation and less chromosomal stability (Baker et al., Adaptation to culture of human embryonic stem cells and oncogenesis in vivo. Nat Biotechnol. 2007;25:207-15). According to the reviewer’s suggestion, we added the experiments using XX hESCs. We could not observe any significant difference between XY and XX lines (data not shown).
COMMRNTS:How did you control for the colony size and resulting size of hEBs? How well? What is the variation in sizes - heterogeneity in size may impact on differentiation. as addressed at Point#5.
REPLY: For the passaging of hESCs, enzymatic and mechanical methods are popularly used. We usually dissect hESC colonies mechanically using customized pipette under microscope (Oh et al., Methods for expansion of human embryonic stem cells. Stem Cells 2005;23:605-9), which makes it possible to maintain homogenously sized colonies. Then, hEBs were filtered through 100-micrometer sieve, and cultured with or without P. This part is now described in Materials and Methods section.
COMMRNTS:It would be of interest to see the effect of progesterone treatment across the time course of treatment- is differentiation being blocked by progesterone or simply delayed?
REPLY: The differentiation of hEBs seemed to be delayed rather than blocked according to our data. The Nestin expression levels of P-treated group caught up those of matched control after 48~72 hours. However, cautious interpretation is necessary since whether this delay may possibly block the later stage development is uncertain at present.
COMMRNTS:Were unused embryos simply the wrong stage or did they arrest in culture? This paragraph of the methods needs clarification.
REPLY: They were discarded because of arrest.
COMMRNTS:Were the animal experiments conducted according to the local ethical committee?
REPLY: All the animal experiments were approved by IACUC of Seoul National University Hospital. The statement is described in the Methods section. The paragraph reads:
Use of human embryonic stem cell lines was approved by Institutional Review Board of Institute of Reproductive Medicine and Population, Medical Research Center, Seoul National University (219932-201307-LR-10-01-1) and animal experiments was approved by Institutional Animal Care and Use Committee (IACUC) of Seoul National University Hospital (15-0016-S1A0) and all the animal experiments were performed according to ethical guideline of Association for Assessment and Accreditation of Laboratory Animal Care (AAALAC).
Round 2
Reviewer 2 Report
The manuscript improved a lot.
I have one minor comment, Since all comparisons in all figures are pairwise, it is more accurate to use asterisk instead of "a" and "b". because this might confuse reader. Subscripts if you are showing difference between all groups
if you are going to use asterisk "*" (keep the line showing the pairwise comparison and put asterisk above it).
Example,
if P< 0.05 use *
if P< 0.01 use **
if P< 0.001 use ***
if you are going to use subscripts "a,b,c" (remove the line showing the pairwise comparison and put subscript above each group bar).
Example,
Figure2 (OCT4)
UN hES bar will have "a"
hEPs_day 5 and hEPs_day 9 bars will have "b"
so this mean UN hES is significantly higher than hEPs_day 5 and hEPs_day 9
Reviewer 3 Report
Thank you for your responses. The manuscript was improved with the additions you have made.
Author Response
Thank you for your kind comments.